# HLA-G Alleles Impact the Perinatal Father–Child HPV Transmission

Nelli T. Suominen [1,2], Michel Roger [3,4], Marie-Claude Faucher [3], Kari J. Syrjänen [5], Seija E. Grénman [1], Stina M. Syrjänen [6,7] and Karolina Louvanto [6,8,9,*]

1 Department of Obstetrics and Gynecology, Turku University Hospital, University of Turku, 20520 Turku, Finland
2 Department of Obstetrics and Gynecology, Vaasa Central Hospital, 65130 Vaasa, Finland
3 Département de Microbiologie et d'Immunologie et Centre de Recherche du Centre Hospitalier de l'Université de Montréal (CHUM), Montreal, QC H2X 0A9, Canada
4 Department of Microbiology, Infectious Diseases and Immunology, University of Montreal, Montreal, QC H3T 1J4, Canada
5 Department of Clinical Research, Biohit Oyj, 00880 Helsinki, Finland
6 Department of Oral Pathology and Radiology, University of Turku, 20520 Turku, Finland
7 Department of Pathology, Turku University Hospital, 20520 Turku, Finland
8 Department of Obstetrics and Gynecology, Faculty of Medicine and Health Technology, Tampere University, 33100 Tampere, Finland
9 Department of Obstetrics and Gynecology, Tampere University Hospital, 33520 Tampere, Finland
* Correspondence: karolina.louvanto@tuni.fi

**Abstract:** The host factors that influence father-to-child human papillomavirus (HPV) transmission remain unknown. This study evaluated whether human leukocyte antigen (HLA)-G alleles are important in father-to-child HPV transmission during the perinatal period. Altogether, 134 father–newborn pairs from the Finnish Family HPV Study were included. Oral, semen and urethral samples from the fathers were collected before the delivery, and oral samples were collected from their offspring at delivery and postpartum on day 3 and during 1-, 2- and 6-month follow-up visits. HLA-G alleles were tested by direct sequencing. Unconditional logistic regression was used to determine the association of the father–child HLA-G allele and genotype concordance with the father–child HPV prevalence and concordance at birth and during follow-up. HLA-G allele G*01:01:03 concordance was associated with the father's urethral and child's oral high-risk (HR)-HPV concordance at birth (OR 17.00, 95% CI: 1.24–232.22). HLA-G allele G*01:04:01 concordance increased the father's oral and child's postpartum oral any- and HR-HPV concordance with an OR value of 7.50 (95% CI: 1.47–38.16) and OR value of 7.78 (95% CI: 1.38–43.85), respectively. There was no association between different HLA-G genotypes and HPV concordance among the father–child pairs at birth or postpartum. To conclude, the HLA-G allele concordance appears to impact the HPV transmission between the father and his offspring.

**Keywords:** HLA-G; human papillomavirus; HPV; transmission; vertical; father

## 1. Introduction

Human papillomavirus (HPV) infection is furthermost known as an accountable factor for the development of cervical cancer in women [1,2]. However, HPV also encompasses a great worldwide burden of diseases in men including medical problems from benign warts to different HPV-related cancers such as anal, penile and oropharyngeal cancers [3]. In children, HPV infection is known to cause benign warts but also juvenile-onset recurrent respiratory papillomatosis (JoRRP), which could be life-threatening if it is left untreated [4,5]. In Europe, the most common HPV genotype is HPV16 followed by HPV31 [6].

Even though HPV infection is one of the most common sexually transmitted infection, non-sexual routes of transmission, such as vertical and horizontal transmission and autoinoculation, have emerged [7–12]. Evidence suggests that newborns can acquire HPV

infection via vertical transmission from the mother or possibly from the father [8,13,14]. Prenatal transmission from the mother to child occurs during pregnancy as an intrauterine transmission [12]. A recent study showed that a mother's persistent HPV16/18 infection is associated also with preterm birth [15]. Perinatal transmission occurs at the time of delivery and immediately thereafter; it results mainly from contact with an infected maternal genital tract [10].

The father's role as a source of HPV infection to the newborn is less known. As HPV DNA has been isolated from vas deferens, seminal fluid, and spermatozoa [16,17], the peri-conceptional transmission (time around fertilization) from the male partner seems at least theoretically possible [10]. Postpartum HPV transmission may occur vertically from either father or mother and horizontally from other caregivers via kissing or digital contact [10].

Even though different modes of transmission have been identified, a child's early-life exposure to HPV infection and genetical and immunological co-factors that facilitate the susceptibility to an infection remain mostly unexplored. Human leukocyte antigen G (HLA-G) is one of those immunological co-factors that is supposed to impact the natural history of HPV as well as other viral infections. In general, HLA-G molecules take part in innate and adaptive immune responses and immune escape in cancer progression [18,19] and infectious diseases [20,21]. HLA-G is classified as a member of the non-classical human leukocyte antigen (HLA) class Ib, since it differs from classical HLA-class I molecules by modulation of the immune response and by having a low degree of polymorphism and restricted tissue distribution [22]. By the expression of placental cells, HLA-G plays a significant role in the maternal–fetal interface [23]. The impact of HLA-G in vertical mother-to-child transmission of viral infection, such as human immunodeficiency virus (HIV) and hepatitis C virus (HCV) infection, has been reported before [24–30]. To date, only one study has evaluated the impact of mother–child HLA-G gene concordance on vertical mother-to-child HPV transmission [31]. No studies on the role of HLA-G in father-to-child HPV transmission have been published. Consequently, our aim was to evaluate whether HLA-G alleles are important in father-to-child HPV transmission during the perinatal period of life. In this study, we determined the association of the father–child HLA-G allele and genotype concordance with the father–child HPV prevalence and concordance at birth and during follow-up.

## 2. Materials and Methods

*The Finnish Family HPV (FFHPV) Study* is a longitudinal cohort follow-up study conducted at Turku University Hospital and University of Turku, Finland. At baseline, a total of 329 families with pregnant women (in their third trimester), fathers-to-be and later their newborn offspring participated in the original study. All families with a minimum of 36 weeks of pregnancy were recruited during their regular obstetrics clinic visit at the hospital. Families were followed up for six years to assess the dynamics of HPV infection between family members as described in detail previously [32,33]. The present study includes 132 fathers and 134 offspring from the FFHPV Study. The cohort represents Caucasian origin, as the Finnish population has the same ethnic background.

*HPV genotyping* Semen, urethral and oral scapings from the fathers and oral scrapings from the newborns were collected for HPV testing with a Cytobrush (MedScand, Malmö, Sweden) as described earlier [13]. Fathers' HPV samples were collected at baseline before delivery. Newborns' oral follow-up samples were collected at delivery and postpartum at day 3 and during 1-, 2- and 6-month follow-up visits. HPV genotyping was performed by the Multimetrix kit (Multimetrix, Progen Biotechnik GmbH, Heidelberg, Germany) identifying 24 different low-risk (LR) and high-risk (HR) HPV genotypes (LR genotypes: 6, 11, 42, 43, 44; and HR genotypes: 16, 18, 26, 31, 33, 35, 39, 45, 51, 52, 53, 56, 58, 59, 66, 68, 70, 73, 82) [34].

*HLA-G typing* DNA from father's and newborn's frozen whole blood samples were extracted for HLA-G typing by using the MagNAPure 96 System (Roche). Determination

of HLA-G alleles was performed by direct DNA sequencing exploring exons 2–4 (1718 bp) of the HLA-G gene regions as described before [35].

STATA SE15.1 (StataCorp, College Station, TX, USA) was used for all statistical analyses. The study cohort included 134 father–child pairs, 132 fathers and their 134 offspring (two pairs of twins), both parties having HLA-G allele determination available. Only those HLA-G alleles and genotypes that were identified both among the fathers and children and were ≥3% prevalent among fathers and/or children were included in the analyses. HLA-G alleles were explored both in high- and low-resolution groups. HLA-G concordance for a specific HLA-G allele was considered if a father and his child both had at least one specific allele; in other words, HLA-G allele concordance was defined if both parties were heterozygous or homozygous for this allele. If one party had an allele and the other did not have the allele, the pair was counted as HLA-G discordant for the allele.

In low-resolution analyses, only one allele group G*01:01+ (including HLA-G alleles G*01:01:01, G*01:01:02, G*01:01:03 and G*01:01:14) was available, while there were four low-resolution genotype groups: 01:01+/01:01+, 01:01+/01:03+, 01:01+/01:04+, and 01:01+/01:06+. In low-resolution analyses, HLA-G allele concordance was divided into four groups by number of shared alleles (1 = discordant for the allele; 2 = 2 shared alleles, 3 = 3 shared alleles, 4 = 4 shared alleles).

In HPV genotyping analyses, HPV species were classified as either LR or HR groups [36]. Multiple-type HPV infections were sorted out as individual HPV genotypes. A child's oral HPV status was counted as the HPV point prevalence (1) at birth covering oral samples at delivery and at day 3 (at discharge) and (2) at postpartum covering samples taken at 1, 2 and 6 months.

In assessing HPV concordance, only those father–child pairs who had an opportunity for HPV infection (i.e., both parties having HPV sample available) were considered. Any-, LR- or HR-HPV concordances were defined if a father and his offspring both were detected positive for any-, LR- or HR-HPV, respectively. HPV prevalence among father–child pairs was calculated as HPV positive (for any-, LR- or HR-HPV) if at least one party, father or child, had at least one positive (any-, LR- or HR-) HPV sample.

Unconditional logistic regression was used to determine the associations (OR) between (1) within father and his child shared HLA-G alleles or genotypes and (2) father–child HPV concordance and prevalence. In high-resolution HLA-G allele analyses, the father–child pairs with both parties being negative for these specific alleles served as a reference, whereas in low-resolution group analyses, HLA-G allele discordant father–child pairs served as a reference. A significance level of 0.05 was used (two-tailed), and 95% confidence intervals (CI) were calculated.

## 3. Results

The mean age of the fathers was 28.8 years (range 19–46 years). Among fathers and their offspring, overall, nine different HLA-G alleles with 19 different genotype combinations were identified. Only those seven alleles and five genotypes that were ≥3% prevalent and were identified in both among the fathers and the children were included in the analyses. The most common HLA-G allele found among fathers and newborns was the wild-type G*01:01:01; 86.4% (*n* = 114) of the fathers, and 85.8% (*n* = 115) of children had the allele. The second most common allele was G*01:01:02; 36.4% (*n* = 48) of the fathers and 32.8% (*n* = 44) of the children had the allele. The most common HLA-G genotype among the fathers and children was G*01:01:01/01:01:01 in 37.1% (*n* = 49) and 36.6% (*n* = 49), respectively, which was followed by G*01:01:01/01:01:02 in 23.5% (*n* = 31) and 21.6% (*n* = 29), respectively. Figure 1 shows allele sharing (Figure 1a) and genotype concordance (Figure 1b) among 134 father–child pairs. The most shared allele was the G*01:01:01, for which 73.9% (*n* = 99) of pairs shared at least two common alleles (i.e., both were at least heterozygous for the allele), which was followed by other alleles that were shared between 2.2% (*n* = 3) and 22.4% (*n* = 30) among the pairs. The most commonly concordant genotype between father–child

pairs was the G*01:01:01/01:01:01; 25.4% (*n* = 34). Overall, 37.3% (*n* = 50) of the father–child pairs had any concordant HLA-G genotype.

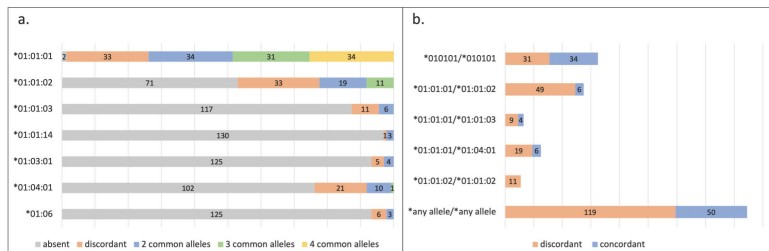

**Figure 1.** Human leukocyte antigen G (HLA-G) (**a**) allele and (**b**) genotype concordance among the 134 father–child pairs from the Finnish Family HPV Study. Stacked bar columns showing the HLA-G (**a**) allele sharing (absent = both missing the allele; discordant = one homo- or heterozygous for the allele and the other absent; 2 common alleles = both heterozygous for the allele; 3 common alleles = one heterozygous for the allele and other homozygous for the allele; 4 common alleles = both homozygous for the allele) and (**b**) genotype concordance (discordance = one having the genotype and the other missing the genotype; concordance = both having the same genotype) between the 134 father–child pairs from the Finnish Family HPV Study. Those HLA-G alleles and genotypes that were ≥3% prevalent among the father–child pairs were included.

When the influence of HLA-G alleles on HPV concordance was evaluated, those father–child pairs who both tested positive for a particular allele (both the father and offspring had at least one allele) were compared to those pairs with discordant HLA-G alleles (only the father or the child had the allele). The father–child pairs with both parties missing the allele served as a reference. Sharing of different alleles was compared to the HPV concordance of father's three anatomical baseline (before birth) HPV status (semen, urethral and oral) and child's oral HPV status at birth and postpartum, as seen in Table 1. HLA-G allele G*01:01:03 concordance was associated with the father's urethral and child's oral HR-HPV concordance at birth (OR 17.00, 95% CI 1.24–232.22). Controversially, during the postpartum period, G*01:01:03 discordance was associated with the father's urethral and child's oral HR-HPV concordance (OR 6.67, 95% CI 1.08–40.97). HLA-G allele G*01:04:01 concordance increased the father's oral and child's postpartum oral any- and HR-HPV concordance with OR 7.50 (95% CI: 1.47–38.16) and OR 7.78 (95% CI: 1.38–43.85), respectively. No association was observed between different HLA-G *genotypes* and HPV concordance among the father–child pairs at birth or during the postpartum period.

The association between father–child HLA-G allele sharing and the common HPV prevalence of fathers and/or children is shown in Table 2. HLA-G allele G*01:01:03 discordance increased the risk of fathers' urethral and/or children's oral any- and HR-HPV prevalence at birth with OR 5.47 (95% CI 1.11–26.89) and 6.76 (95% CI 1.37–33.35), whereas G*01:01:03 concordance increased the risk of the fathers' oral and/or children's oral HR-HPV prevalence at birth with OR 8.81 (95% CI 1.00–77.94). G*01:06 allele concordance increased the LR-HPV prevalence of fathers' semen/urethral and/or children's oral sites at birth; OR ranged between 12.00 (95% CI 1.03–140.19) and 16.15 (95% CI 1.37–190.72). No association was seen between HLA-G allele sharing and the fathers' and/or children's postpartum HPV prevalence, as shown in Table S1 (see Supplementary Materials).

Table 3 summarizes the association of father–child HLA-G genotype concordance, in both high- and low-resolution groups, with the HPV prevalence of the fathers and/or children. No significant association was found in high-resolution group, whereas in the low-resolution group, G*01:01/01:04 concordance was associated with the greater HR-HPV prevalence of fathers' semen and/or children's oral sites during the postpartum period, showing an OR value of 7.00 (95% CI 1.14–42.97). Similarly, G*01:01/01:04 concordance increased the fathers' any urethral and/or children's postpartum any oral HPV prevalence with an OR value of 6.50 (95% CI 1.05–40.13).

**Table 1.** Association of human leukocyte antigen G (HLA-G) allele sharing with human papillomavirus (HPV) concordance among 134 father–child pairs of the Finnish Family HPV Study. The father–child pairs with both parties being negative for these specific alleles served as a reference in high-resolution HLA-allele groups. In low-resolution groups, discordant father–child pairs served as a reference.

| | | Children Oral HPV [b] OR (95% CI) | | | | | |
| | | Father Semen [b] | | Father Urethral [b] | | Father Oral [b] | |
| HLA-G Allele Concordance [a] | Allele Sharing | Any | HR | Any | HR | Any | HR |
|---|---|---|---|---|---|---|---|
| **At birth** | | | | | | | |
| 01:01:02 | 0 | 1.00 | 1.00 | 1.00 | 1.00 | 1.00 | 1.00 |
| | 1 | 0.27 (0.03-2.53) | NA | 0.39 (0.07-2.15) | 0.32 (0.03-3.14) | 1.18 (0.28-4.96) | 1.04 (0.21-5.20) |
| | ≥2 | 0.36 (0.04-3.52) | 0.43 (0.04-4.26) | 1.18 (0.24-5.70) | 0.59 (0.06-6.18) | NA | NA |
| 01:01:03 | 0 | 1.00 | 1.00 | 1.00 | 1.00 | 1.00 | 1.00 |
| | 1 | 2.22 (0.20-25.00) | NA | 0.60 (0.07-5.54) | NA | NA | NA |
| | ≥2 | NA | NA | 8.44 (0.69-103.70) | **17.00 (1.24-232.22)** | 1.08 (0.11-10.89) | 1.18 (0.11-12.21) |
| 01:04:01 | 0 | 1.00 | 1.00 | 1.00 | 1.00 | 1.00 | 1.00 |
| | 1 | 2.67 (0.23-30.80) | 8.50 (0.44-163.89) | 0.46 (0.05-4.15) | 1.06 (0.10-10.84) | 0.52 (0.06-4.74) | 0.56 (0.06-5.39) |
| | ≥2 | 2.67 (0.23-30.80) | 2.83 (0.24-34.14) | 3.70 (0.21-64.51) | NA | 0.83 (0.08-8.08) | 1.13 (0.11-11.95) |
| 01:06 | 0 | 1.00 | 1.00 | 1.00 | 1.00 | 1.00 | 1.00 |
| | 1 | NA | NA | NA | NA | NA | NA |
| | ≥2 | 3.58 (0.28-45.80) | 3.50 (0.27-46.05) | 3.91 (0.23-67.57) | NA | 4.89 (0.28-85.63) | NA |
| **Low resolution** | | | | | | | |
| 01:01+ | 1 | 1.00 | 1.00 | 1.00 | 1.00 | 1.00 | 1.00 |
| | 2 | NA | NA | 0.33 (0.01-11.94) | NA | NA | NA |
| | 3 | NA | NA | 0.18 (0.01-4.26) | 0.10 (0.003-3.15) | NA | NA |
| | 4 | NA | NA | 0.26 (0.01-4.59) | 0.14 (0.01-2.67) | NA | NA |
| **Postpartum** | | | | | | | |
| 01:01:02 | 0 | 1.00 | 1.00 | 1.00 | 1.00 | 1.00 | 1.00 |
| | 1 | 0.83 (0.20-3.54) | 1.60 (0.33-7.65) | 1.55 (0.40-5.99) | 2.50 (0.56-11.21) | 1.63 (0.41-6.51) | 1.92 (0.46-7.95) |
| | ≥2 | 0.15 (0.02-1.39) | 0.32 (0.03-3.15) | 0.58 (0.10-3.23) | NA | 0.40 (0.04-3.79) | 0.51 (0.05-5.00) |

**Table 1.** *Cont.*

| | | Children Oral HPV [b] OR (95% CI) | | | | | |
|---|---|---|---|---|---|---|---|
| | | Father Semen [b] | | Father Urethral [b] | | Father Oral [b] | |
| HLA-G Allele Concordance [a] | Allele Sharing | Any | HR | Any | HR | Any | HR |
| 01:01:03 | 0 | 1.00 | 1.00 | 1.00 | 1.00 | 1.00 | 1.00 |
| | 1 | 3.90 (0.49-31.20) | 3.88 (0.47-31.91) | 4.90 (0.86-27.88) | **6.67 (1.08-40.97)** | NA | NA |
| | ≥2 | NA | NA | NA | NA | NA | NA |
| 01:04:01 | 0 | 1.00 | 1.00 | 1.00 | 1.00 | 1.00 | 1.00 |
| | 1 | 1.54 (0.14-16.80) | NA | 2.25 (0.36-14.03) | 3.08 (0.46-20.70) | 1.25 (0.13-12.24) | 0.97 (0.10-9.57) |
| | ≥2 | 6.94 (0.99-48.55) | 6.43 (0.90-46.06) | 0.75 (0.08-6.94) | 1.23 (0.12-12.47) | **7.50 (1.47-38.16)** | **7.78 (1.38-43.85)** |
| 01:06 | 0 | 1.00 | 1.00 | 1.00 | 1.00 | 1.00 | 1.00 |
| | 1 | NA | NA | 4.91 (0.28-84.58) | NA | NA | NA |
| | ≥2 | 3.55 (0.20-61.38) | 3.56 (0.20-62.63) | NA | NA | NA | NA |
| Low resolution | | | | | | | |
| 01:01+ | 1 | 1.00 | 1.00 | 1.00 | 1.00 | 1.00 | 1.00 |
| | 2 | NA | NA | 0.13 (0.004-4.00) | NA | NA | NA |
| | 3 | NA | NA | 0.57 (0.03-11.85) | 0.33 (0.01-8.18) | NA | NA |
| | 4 | NA | NA | 0.18 (0.01-3.22) | 0.18 (0.01-3.32) | NA | NA |

[a] Only those alleles that were ≥3% prevalent were included in the analyses. [b] Father's semen, urethral or oral HPV at baseline and child's oral HPV at birth and postpartum. 0 = no shared alleles; 1 = discordant for the allele; ≥2 = at least 2 shared alleles, i.e., both heterozygous or homozygous for the allele; 2 = 2 shared alleles, 3 = 3 shared alleles, 4 = 4 shared alleles. LR-HPV groups as HLA-G alleles 01:01:01, 01:01:14 and 01:03:01 were taken out due to small number of cases. Statistically significant OR:s (*p*-value ≤ 0.05) are shown in bold. NA = not applicable.

**Table 2.** Association of HLA-G allele sharing with HPV prevalence at birth among 134 father–child pairs of the Finnish Family HPV Study. The father–child pairs with both parties being negative for these specific alleles served as a reference in high-resolution HLA allele groups. In low-resolution HLA-G allele groups, discordant father–child pairs served as a reference.

| HLA-G Allele Concordance [a] | Allele Sharing | HPV: Father Semen/Child Oral [b] OR (95% CI) | | | HPV: Farther Urethral/Child Oral [b] OR (95% CI) | | | HPV: Father Oral/Child Oral [b] OR (95% CI) | | |
|---|---|---|---|---|---|---|---|---|---|---|
| | | Any | LR | HR | Any | LR | HR | Any | LR | HR |
| | 0 | 1.00 | 1.00 | 1.00 | 1.00 | 1.00 | 1.00 | 1.00 | 1.00 | 1.00 |
| *01:01:01 | 1 | 0.60 (0.03-10.51) | NA | 0.45 (0.03-8.02) | 0.68 (0.04-11.95) | NA | 0.63 (0.04-11.08) | 0.68 (0.04-11.95) | NA | 0.60 (0.03-10.51) |
| | ≥2 | 1.06 (0.06-17.47) | NA | 0.81 (0.05-13.40) | 0.88 (0.05-14.55) | NA | 0.71 (0.04-11.76) | 0.75 (0.05-12.34) | NA | 0.58 (0.04-9.57) |
| | 0 | 1.00 | 1.00 | 1.00 | 1.00 | 1.00 | 1.00 | 1.00 | 1.00 | 1.00 |
| *01:01:02 | 1 | 1.46 (0.64-3.35) | 1.32 (0.39-4.44) | 1.41 (0.61-3.25) | 1.66 (0.72-3.85) | 0.81 (0.23-2.83) | 1.87 (0.80-4.38) | 1.64 (0.71-3.80) | 1.34 (0.36-4.98) | 1.62 (0.70-3.79) |
| | ≥2 | 1.14 (0.48-2.70) | 1.56 (0.46-5.30) | 0.92 (0.38-2.23) | 0.79 (0.33-1.91) | 0.44 (0.09-2.14) | 0.92 (0.37-2.29) | 0.76 (0.31-1.88) | 0.31 (0.04-2.65) | 0.83 (0.33-2.09) |
| | 0 | 1.00 | 1.00 | 1.00 | 1.00 | 1.00 | 1.00 | 1.00 | 1.00 | 1.00 |
| *01:01:03 | 1 | 1.09 (0.30-3.96) | 3.03 (0.68-13.45) | 1.47 (0.40-5.35) | **5.47 (1.11-26.89)** | 0.73 (0.09-6.15) | **6.76 (1.37-33.35)** | 0.61 (0.15-2.47) | 1.02 (0.12-8.83) | 0.44 (0.09-2.17) |
| | ≥2 | 1.09 (0.21-5.62) | NA | 1.47 (0.28-7.59) | 1.37 (0.26-7.06) | NA | 1.69 (0.33-8.76) | 7.08 (0.80-62.57) | NA | **8.81 (1.00-77.94)** |
| | 0 | 1.00 | 1.00 | 1.00 | 1.00 | 1.00 | 1.00 | 1.00 | 1.00 | 1.00 |
| *01:01:14 | 1 | NA | NA | NA | NA | NA | NA | NA | NA | NA |
| | ≥2 | 0.54 (0.05-6.11) | 2.94 (0.25-34.25) | NA | 2.41 (0.21-27.30) | 3.60 (0.31-42.16) | 1.42 (0.09-23.27) | 0.66 (0.06-7.51) | 5.09 (0.43-60.73) | NA |
| | 0 | 1.00 | 1.00 | 1.00 | 1.00 | 1.00 | 1.00 | 1.00 | 1.00 | 1.00 |
| *01:03:01 | 1 | 0.69 (0.11-4.26) | NA | 0.91 (0.15-5.64) | 0.80 (0.13-4.94) | NA | 0.97 (0.16-6.00) | 0.88 (0.14-5.46) | NA | 1.12 (0.18-6.93) |
| | ≥2 | 0.34 (0.03-3.40) | NA | 0.46 (0.05-4.50) | 1.20 (0.16-8.77) | NA | 1.45 (0.20-10.64) | 0.44 (0.04-4.35) | NA | 0.56 (0.06-5.52) |
| | 0 | 1.00 | 1.00 | 1.00 | 1.00 | 1.00 | 1.00 | 1.00 | 1.00 | 1.00 |
| *01:04:01 | 1 | 0.60 (0.23-1.58) | 0.76 (0.16-3.70) | 0.51 (0.18-1.42) | 0.90 (0.34-2.37) | 0.34 (0.04-2.76) | 1.09 (0.41-2.87) | 1.20 (0.46-3.15) | NA | 1.54 (0.58-4.08) |
| | ≥2 | 0.82 (0.23-2.85) | 0.59 (0.07-5.03) | 1.06 (0.30-3.70) | 0.41 (0.10-1.65) | 1.62 (0.31-8.46) | 0.30 (0.06-1.44) | 1.76 (0.50-6.14) | 1.93 (0.37-10.23) | 1.57 (0.45-5.52) |
| | 0 | 1.00 | 1.00 | 1.00 | 1.00 | 1.00 | 1.00 | 1.00 | 1.00 | 1.00 |
| *01:06 | 1 | 0.21 (0.02-1.88) | NA | 0.28 (0.03-2.49) | 0.60 (0.11-3.39) | 1.62 (0.17-14.92) | 0.28 (0.03-2.47) | 0.26 (0.03-2.33) | NA | 0.33 (0.04-2.96) |
| | ≥2 | NA | **12.00 (1.03-140.19)** | NA | 2.39 (0.21-27.09) | **16.15 (1.37-190.72)** | 2.80 (0.25-31.73) | 2.64 (0.23-29.91) | 4.86 (0.41-58.04) | 3.35 (0.30-37.95) |
| Low resolution | | | | | | | | | | |
| | 1 | 1.00 | 1.00 | 1.00 | 1.00 | 1.00 | 1.00 | 1.00 | 1.00 | 1.00 |
| *01:01+ | 2 | 3.43 (0.29-40.95) | NA | 3.43 (0.29-40.95) | 0.50 (0.05-4.67) | NA | 0.36 (0.04-3.52) | 3.43 (0.29-40.95) | NA | 2.63 (0.22-31.35) |
| | 3 | 2.29 (0.21-24.68) | NA | 1.50 (0.14-16.32) | 0.81 (0.10-6.58) | NA | 0.71 (0.09-5.73) | 2.44 (0.23-26.30) | NA | 2.44 (0.23-26.30) |
| | 4 | 3.00 (0.30-30.02) | NA | 2.30 (0.23-23.02) | 0.91 (0.12-6.76) | NA | 0.76 (0.10-5.67) | 2.04 (0.20-20.44) | NA | 1.50 (0.15-15.08) |

[a] Only those alleles that were ≥3% prevalent were included in the analyses. [b] Father's semen, urethral or oral HPV prevalence at baseline and/or child's oral HPV prevalence at birth. 0 = no shared alleles; 1 = discordant for the allele; ≥2 = at least 2 shared alleles i.e., both heterozygous or homozygous for the allele; 2 = 2 shared alleles, 3 = 3 shared alleles, 4 = 4 shared alleles. Statistically significant OR:s ($p$-value ≤ 0.05) are shown in bold. NA = not applicable.

**Table 3.** Association of HLA-G genotype concordance with HPV prevalence at birth and postpartum among 134 father–child pairs of the Finnish Family HPV Study. HLA-G genotype discordant father–child pairs served as a reference.

| HLA-G Genotype Concordance [a] | Children Oral HPV [b] OR (95% CI) | | | | | | | | |
| --- | --- | --- | --- | --- | --- | --- | --- | --- | --- |
| | Father Semen [b] | | | Father Urethral [b] | | | Father Oral [b] | | |
| | Any | LR | HR | Any | LR | HR | Any | LR | HR |
| **At Birth** | | | | | | | | | |
| 01:01:01/01:01:01 | 0.84 (0.32-2.24) | 1.93 (0.32-11.43) | 0.90 (0.33-2.44) | 0.75 (0.28-1.99) | 2.59 (0.46-14.46) | 0.79 (0.29-2.16) | 0.98 (0.36-2.67) | 1.79 (0.30-10.61) | 0.87 (0.31-2.43) |
| 01:01:01/01:01:02 | 1.84 (0.31-11.01) | 0.90 (0.09-8.80) | 1.29 (0.24-7.03) | 0.18 (0.02-1.62) | NA | 0.20 (0.02-1.85) | 0.23 (0.02-2.10) | NA | 0.29 (0.03-2.73) |
| 01:01:01/01:01:03 | 3.75 (0.27-51.37) | NA | 3.75 (0.27-51.37) | 1.50 (0.11-21.31) | NA | 1.50 (0.11-21.31) | NA | NA | NA |
| 01:01:01/01:04:01 | 1.11 (0.18-6.97) | 1.20 (0.09-16.44) | 1.71 (0.27-10.92) | 0.63 (0.09-4.33) | 8.00 (0.57-111.96) | 0.25 (0.02-2.59) | 1.00 (0.16-6.35) | NA | 0.50 (0.07-3.45) |
| low resolution | | | | | | | | | |
| 01:01+/01:01+ | 1.31 (0.56-3.03) | 1.31 (0.34-5.09) | 1.53 (0.64-3.67) | 1.12 (0.48-2.61) | 1.10 (0.28-4.37) | 1.08 (0.46-2.55) | 0.84 (0.36-1.96) | NA | 0.62 (0.26-1.46) |
| 01:01+/01:03+ | 2.50 (0.10-62.60) | . . . | 2.50 (0.10-62.60) | 1.33 (0.06-31.12) | . . . | 1.33 (0.06-31.12) | 2.50 (0.10-62.60) | . . . | 2.50 (0.10-62.60) |
| 01:01+/01:04+ | 1.07 (0.22-5.14) | 0.81 (0.06-10.48) | 1.60 (0.32-7.90) | 0.35 (0.06-2.12) | 5.14 (0.40-66.15) | 0.15 (0.02-1.46) | 1.25 (0.26-6.07) | NA | 0.80 (0.16-3.88) |
| 01:01+/01:06+ | NA | NA | NA | 3.00 (0.15-59.89) | 8.00 (0.31-206.37) | 8.00 (0.31-206.37) | 8.00 (0.31-206.37) | NA | 8.00 (0.31-206.37) |
| **Postpartum** | | | | | | | | | |
| 01:01:01/01:01:01 | 1.20 (0.45-3.18) | 0.80 (0.19-3.35) | 1.29 (0.48-3.44) | 1.07 (0.40-2.82) | 1.03 (0.28-3.80) | 1.23 (0.46-3.28) | 0.95 (0.36-2.51) | 0.71 (0.17-2.96) | 1.09 (0.41-2.92) |
| 01:01:01/01:01:02 | NA | 0.48 (0.05-4.49) | 7.11 (0.77-65.80) | 0.68 (0.12-3.73) | NA | 1.05 (0.19-5.74) | 0.88 (0.16-4.82) | NA | 1.47 (0.27-8.09) |
| 01:01:01/01:01:03 | 0.67 (0.05-9.47) | 1.17 (0.07-18.35) | 0.67 (0.05-9.47) | 3.75 (0.27-51.37) | NA | 3.75 (0.27-51.37) | 3.75 (0.27-51.37) | NA | 3.75 (0.27-51.37) |
| 01:01:01/01:04:01 | 2.75 (0.40-18.88) | NA | 3.43 (0.49-23.77) | 3.14 (0.45-21.96) | 1.75 (0.12-24.65) | 1.57 (0.24-10.09) | 2.50 (0.36-17.32) | 5.00 (0.24-104.15) | 1.25 (0.20-7.96) |
| low resolution | | | | | | | | | |
| 01:01+/01:01+ | 1.32 (0.57-3.04) | 0.99 (0.35-2.87) | 1.90 (0.79-4.56) | 1.80 (0.77-4.23) | 0.99 (0.32-3.06) | 1.67 (0.69-4.06) | 1.02 (0.44-2.38) | 1.45 (0.38-5.56) | 1.01 (0.43-2.38) |
| 01:01+/01:03+ | NA | NA | 1.33 (0.06-31.12) | NA | NA | 0.75 (0.03-17.51) | NA | NA | 1.33 (0.06-31.12) |
| 01:01+/01:04+ | 5.69 (0.94-34.46) | NA | **7.00 (1.14-42.97)** | **6.50 (1.05-40.13)** | 1.14 (0.09-14.78) | 3.71 (0.70-19.59) | 4.28 (0.71-25.92) | 4.50 (0.23-88.24) | 2.44 (0.47-12.63) |
| 01:01+/01:06+ | 3.00 (0.15-59.89) | 2.00 (0.08-51.59) | 8.00 (0.31-206.37) | 0.75 (0.04-14.97) | 0.75 (0.04-14.97) | 2.00 (0.08-51.59) | 3.00 (0.15-59.89) | 2.00 (0.08-51.59) | 2.00 (0.08-51.59) |

[a] Only those genotypes that were ≥3% prevalent were included in the analyses. [b] Father's semen, urethral or oral HPV prevalence at baseline and/or child's oral HPV prevalence at birth and postpartum. Genotype 01:01:02/01:01:02 was taken out due to the small number of cases. Statistically significant OR:s (*p*-value ≤ 0.05) are shown in bold. NA = not applicable.

## 4. Discussion

The acquisition of HPV infection in early life is supposed to be facilitated by many complex immunological, genetical and epigenetical co-factors [37]. The father's role in a child's early life exposure to HPV infection remains unclear. According to our results, certain HLA-G alleles appear to have an impact on father–child HPV concordance and prevalence in the perinatal period. However, HLA-G genotypes did not show any influence on the HPV concordance among the father–child pairs. Only a few other studies have evaluated the father's role in perinatal HPV infection with controversial results [13,14,38]. The study with the Polish family cohort (146 parental couples and their newborns) showed that a father's oral HPV16/18 infection increased the newborn's oral HPV16/18 infection at birth [14]. With our Finnish Family cohort, Rintala and colleagues showed that the newborn's genital HPV positivity associated with the mother's oral HR-HPV detected before delivery, whereas the newborn's oral HPV positivity at the age of six months was associated with father's oral HR-HPV detected before delivery [13]. In both studies, type-specific concordance was not analyzed. In addition, another recent study with our Finnish Family cohort showed that the incident oral HR-HPV infection for the child was predicted by the HR-HPV seropositivity of the father [39]. In contrast, Smith and colleagues showed HPV transmission from parents to newborns to be rare in an American population in Iowa, as only one of the 574 mother–child pairs and none of the 68 father–child pairs had a concordant HPV type [38].

The role of HLA-G in vertical HPV transmission is even less studied. To our knowledge, there is only one previous study with the same Finnish Family HPV cohort that has evaluated the HLA-G and vertical mother-to-child HPV transmission [31]. In that study, HLA-G allele or genotype concordance did not show any impact on mother-to-child genotype-specific HPV transmission [31]. To date, no studies on HLA-G in father–child HPV concordance or transmission have been reported.

HLA-G in vertical HIV transmission is more studied than HLA-G in vertical HPV transmission. Several studies have shown HLA-G polymorphism to influence the risk of mother-to-child HIV transmission [24–29]. With regard to mother–child HLA-G concordance, the data are controversial and sparse. One small prospective cohort study (N = 34) run in New York City suggested that mother–child discordance in exon 2 is associated with a reduced risk of perinatal HIV infection [24]. However, two larger studies have not found any association between HLA-G allele concordance and vertical mother-to-child HIV transmission [25,29].

One study has shown an association of the HLA-G*01:04:01 allele with vertical mother-to-child HCV transmission; G*01:04:01 was significantly higher in HCV-positive children than HCV-negative children (22.5% vs. 6.8%) [30]. Interestingly, we found HLA-G*01:04:01 allele concordance to increase the father's oral and child's postpartum oral any-and HR-HPV concordance. According to the literature, the data are partly inconsistent, and no consistency of specific risk alleles for the vertical transmission of viral infections exist. To the best of our knowledge, no studies with HLA-G and father-to-child transmission of HIV nor other viruses have been published before.

In this study with a cohort of fathers and their newborns, we identified only nine different HLA-G alleles. To date, overall, 117 HLA-G alleles have been identified (IPD-IMGT/HLA Database, July 2023) [40]. The relatively low number of different HLA-G alleles we found was not surprising considering that the Finnish population has a quite restricted and homogenous gene pool due to historical isolation. The most common allele observed among both fathers and their offspring in our cohort was the wild-type G*01:01:01, which was expected as G*01:01:01 is usually the most common HLA-G allele in all population studies so far [41].

We showed HLA-G concordance with certain specific alleles to have an impact on father–child HPV concordance (in any-, LR- and HR-HPV groups), but whether it indicates HPV transmission from father to child is questionable. We showed that the type-specific father–child HPV concordance was associated with HLA-G allele G*01:04:01. HLA-

G*01:04:01 father–child concordance was related to the father's oral and child's postpartum oral any- and HR-HPV concordance. In this case, HPV type-specific concordance was seen in two father–child pairs with HR-HPV genotypes 33 and 70 (50% of concordant pairs had HR-HPV genotype specific concordance. Further evaluations showed that if the mother's oral HPV status were also taken into account, the adjusted OR values for G*01:04:01 father–child concordance and the father's oral and child's postpartum oral any- and HR-HPV concordance remained statistically significant; adjusted OR values: 11.50 (95% CI 1.77–74.87) and 9.86 (95% CI 1.49–65.12), respectively. This finding is suggestive for vertical father-to-child HPV transmission in this case. Interestingly, in the study with the same Finnish Family cohort including mother–child pairs, discordant mother–child HLA-G allele G*01:04:01 increased the risk of the child's oral LR-HPV infection at birth [31].

According to our results, HLA-G*01:01:03 allele father–child concordance relates to the father's urethral and child's oral HR-HPV concordance at birth. Controversially, when a child's oral HPV status was determined postpartum, an association between G*01:01:03 allele discordance and the father's urethral and child's oral HR-HPV concordance was seen. Moreover, HLA-G allele G*01:01:03 discordance seems to increase the risk of the father's urethral and/or child's oral any- and HR-HPV positivity at birth, whereas G*01:01:03 concordance was associated with the higher risk of father's oral and/or child's oral HR-HPV positivity at birth. This contradiction remains unexplainable assuming that biologically HLA-G allele discordance is supposed to reduce the risk of infection at any anatomical site. Therefore, we investigated separately the fathers' HPV prevalence at baseline (before birth) and children's HPV prevalence at birth; G*01:01:03 allele discordance seemed to increase solely the father's urethral any- and HR-HPV positivity but not the child's oral HPV positivity at birth. G*01:01:03 allele concordance lost statistical significance when the fathers' oral HR-HPV prevalence and children's oral HR-HPV prevalence at birth were explored separately; thus, it did not show an impact on one or the other's oral HR-HPV prevalence alone. Interestingly, in our recently published study, we found the presence of men's allele G*01:01:03 to associate with an increased risk for urethral HR-HPV infections [42].

Allele G*01:06 concordance was associated with the LR-HPV prevalence of fathers' semen and urethral and/or children's oral HPV at birth. When we explored this association by fathers' and children's prevalence alone, G*01:06 concordance was not associated separately with the father's semen or child's oral LR-HPV positivity at birth. However, G*01:06 concordance increased the father's urethral LR-HPV positivity but not the child's oral LR-HPV positivity at birth, when the prevalence of fathers and children was analyzed separately. Interestingly, the study of the Finnish Family cohort with mother–child pairs showed that G*01:06 mother–child discordance increased the child's oral LR-HPV positivity but had no impact on the mother's site [31].

The main limitation of this study is that we could not stratify analyses by HPV genotype due to the relatively low sample size of fathers and newborns. A small sample size of fathers and children with uncommon HLA-G alleles may reduce the detection rate of significant associations between father–child HLA-G allele concordance and father–child HPV concordance. In addition, especially for the perinatal period, newborns might be also exposed to HPV from siblings and other caregivers. These covariables were not considered in this study. In addition, as the study cohort represents the Finnish population, which has a restricted gene pool, the findings of this study may not be generalized to other populations.

For many, it is still questionable whether a child's HPV status at birth represents passive HPV contamination or a true infection. However, a recently published study with the FFHPV cohort showed that part of the newborns born to seronegative mothers showed seroconversion to HPV6, HPV11, HPV16 and HPV18 recorded after birth [43]. According to this finding, there is a reason to suggest that newborns had acquired HPV infection somewhere in their body as they had created an immune response for HPV already in early infancy. In fact, our previous findings by Koskimaa et al. showed that

an HPV16-specific immune response exists among these unvaccinated and sexually-naïve children [44–46]. The oral infection with LR HPV6 and HPV11 types is known to cause juvenile-onset recurrent respiratory papillomatosis [4]. Even if the JoRRP is rare, and the lesions it causes are benign, recurrent disease may need repeated surgery and can persist into adulthood [4]. Given that, the consequences of newborn's exposure for HPV infection should not be ignored. Furthermore, the risk of genital precancerous lesions in adolescents and young adults based on vertical transmission is not fully understood. Better knowledge of the natural history of HPV in early childhood is crucial to create the most effective preventive strategies for HPV infection-related diseases as to determine the optimal timing of the prophylactic HPV vaccination.

## 5. Conclusions

This is the first study to our knowledge to evaluate the association of the father–child HLA-G allele and genotype concordance with the father–child HPV prevalence and concordance at birth and during the perinatal period. According to our results, father–child concordance of certain HLA-G alleles appears to have an impact on father–child HPV concordance and prevalence in the perinatal period. However, father–child HLA-G genotype concordance did not show such an impact. Due to a relatively low sample size, we could not evaluate the results by genotype-specific HPV concordance. Thus, the results are approximate and should be further validated by larger studies, with HLA-G and genotype-specific HPV data, in the future.

To conclude, father–child HLA-G concordance might play some role in father–child HPV concordance and prevalence at birth and during the perinatal period of life. The regulatory role of HLA-G in a child's susceptibility to HPV infection and the role of the father in the transmission chain need further investigations.

**Supplementary Materials:** Table S1: Association of Human leukocyte antigen G (HLA-G) allele sharing with postpartum Human papillomavirus (HPV) prevalence among 134 father-child pairs of the Finnish Family HPV study. The following supporting information can be downloaded at: https://www.mdpi.com/article/10.3390/cimb45070366/s1.

**Author Contributions:** S.M.S. and S.E.G. conceived and designed the study. S.M.S., M.R. and M.-C.F. performed the HPV genotyping and HLA-G testing. S.M.S., K.J.S. and K.L. collated the data. N.T.S., K.L., S.M.S. and K.J.S. interpreted the data. N.T.S. and K.L. conducted the statistical analyses. N.T.S. wrote the first draft of the manuscript, and all authors commented on previous versions of the manuscript. K.L. is the guarantor. The corresponding author attests that all listed authors meet authorship criteria and that no others meeting the criteria have been omitted. All authors have read and agreed to the published version of the manuscript.

**Funding:** This study was funded by the Sigrid Jusélius Foundation, the Finnish Medical Foundation, the Academy of Finland, the Finnish Cancer Foundation, the Päivikki and Sakari Sohlberg Foundation, and the Government Special Foundation (EVO) to Turku University Hospital.

**Informed Consent Statement:** The study was performed in line with the principles of the Declaration of Helsinki. The study protocol and its amendment (#3/1998, #2/2006 and 45/180/2010) have been approved by the Ethics Committee of Turku University Hospital. Written informed consent to participate was obtained from all adult participants. Written informed consent for child's participation was obtained from both parents of the child.

**Data Availability Statement:** The datasets analyzed during the current study are available from the guarantor Karolina Louvanto on reasonable request.

**Acknowledgments:** This study has been supported by the Sigrid Jusélius Foundation, the Finnish Medical Foundation, the Academy of Finland, the Finnish Cancer Foundation, the Päivikki and Sakari Sohlberg Foundation, and the Government Special Foundation (EVO) to Turku University Hospital. The skillful technical assistance of Tatjana Peskova, Mariia Henttinen and Keitlin Adel and review assistance of Claudie Laprise are gratefully acknowledged.

**Conflicts of Interest:** The authors declare no conflict of interest.

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
