# Peer review of "HLA-G Alleles Impact the Perinatal Father–Child HPV Transmission"

_cimb, doi:10.3390/cimb45070366_

Round 1

Reviewer 1 Report

Please read the attachment. Thank you.

The technical English is generally clear and understandable. The manuscript effectively describes the study design, methodology, and statistical analysis techniques.

Author Response

General Comments

The manuscript investigates the role of HLA-G alleles in father-to-child HPV transmission. The study utilizes samples collected from father-newborn pairs in the Finnish Family HPV Study and employs direct sequencing to test for HLA-G alleles. The authors use unconditional logistic regression to analyze the association between father-child HLA-G allele concordance and HPV prevalence and concordance at birth and during follow-up. In short, the study provides valuable insights into the impact of HLA-G allele concordance on HPV transmission between fathers and their offspring.

We thank the reviewer#1 for the taking time to review our paper and giving excellent points to improve our paper forward. All changes are flagged in the manuscript file with highlighted yellow text. A clean version without markings has also been uploaded to the submission system.

Specific Comments

  • ⎯  Introduction: please add a paragraph to introduce the outline of the manuscript.

The introduction of the outline of the manuscript have been added to the end of the introduction section.

  • ⎯  Please remove subsection 2.1 Statistical Analyses because section 2. Materials and Methods has only one subsection. Therefore, it is not necessary.

The subsection 2.1 has been removed.

  • ⎯  Line 75: there are the same noun phrases (continuously) of "Finnish Family HPV Study"; please remove the one.

This has been now corrected and removed.

  • ⎯  Lines 133-134: please check the grammar and structure of the sentences.

This has been now corrected.

  • ⎯  Section 5. Conclusion: please extend this section. Firstly, please provide a summary of this study process. Then, present the main findings of this study, the limitations, and the improvement of the proposed methodology.

Thank you for this comment. The conclusion section has been now extended as recommended.

  • ⎯  Please format the manuscript following the guidance of the journal template.

The manuscript should be now according to the journal template.

Constructive Questions:

1) Can you provide more information on potential confounding variables considered in the statistical analysis and the methods used for adjustment? Addressing potential confounders is essential to ensure the robustness of the observed associations between HLA-G allele concordance and HPV transmission.

This study was solely designed to evaluate the different HLA-G allele and genotype role in the risk for HPV prevalence or concordance between the father and the newborn. HPV prevalence and the natural history of HPV as the confounding factors for the fathers and the children have been previously reported in detailed (Rintala et al 2002, Rintala et al 2005, Rintala et al 2005b; Syrjänen et al 2021). For HLA-G we used the low and high resolution of the alleles to truly investigate the role of the HLA-G among father and child HPV transmission. In assessing HPV concordance only those father-child pairs who had an opportunity for HPV infection (i.e. both parties having HPV sample available) were considered and only HLA-G alleles that were more than 3% prevalent among fathers and children were included to the analyses. All this information is in the statistical analyses part of the manuscript.

2) Considering the wide confidence intervals observed for some odds ratios (ORs), can you discuss the possible sources of uncertainty or variability in the data? It would be beneficial to explore factors that may have contributed to the wide range of estimates and discuss the implications for the reliability and precision of the findings.

The wide confidence intervals observed for some odds rations is due to relative low sample size. Thus, the results are approximate as stated at the discussion and conclusion sections.

3) Given that the study is based on the Finnish Family HPV Study population, which may not be representative of other populations, can you discuss the potential limitations and generalizability of the findings? It would be helpful to address how the genetic background and HPV prevalence in different populations may impact the interpretation and applicability of the results.

Thank you for this excellent point. We do point out in the discussion on how the relatively low number of different HLA-G alleles we found was not surprising considering that Finnish population has a quite restricted and homogenous gene pool due to historical isolation. We now added to the discussion a part to about generalizability as it is true that results may vary with different gene pools.

In conclusion, this study provides valuable insights into the influence of HLA-G alleles on father-to-child HPV transmission during the perinatal period. The findings contribute to our understanding of host factors involved in HPV transmission and have implications for further research and potential clinical applications. Addressing the comments above would strengthen the manuscript and improve its overall quality.

Thank you for this kind comment. We agree that this will bring important information for future research implications. Please see responses to all the comments above.

Reviewer 2 Report

Suominen and colleagues have presented a well-structured article entitled “HLA-G alleles impact the perinatal father-child HPV transmission”. The article is quite interesting and presented a new concept related to the role of HLA-G alleles in vertical father-to-child HPV transmission. I only have minor comments for the authors:

1. The impact of HLA-G alleles in the vertical transmission of several viral infections including HIV and HCV has been reported in many previous studies. Authors should mention that in their introduction. Additionally, I recommend discussing if there is a common HLA-G feature that contributes to the vertical transmission of different viruses by comparing the current study findings with other studied viruses’ reports.

2. I recommend adding an introductory sentence in the first paragraph of the discussion instead of starting the discussion with (According to our results, .....).

Minor editing of English language required

Author Response

Comments and Suggestions for Authors

Suominen and colleagues have presented a well-structured article entitled “HLA-G alleles impact the perinatal father-child HPV transmission”. The article is quite interesting and presented a new concept related to the role of HLA-G alleles in vertical father-to-child HPV transmission.

We thank the reviewer#2 for the taking time to review our paper and for the good comments.

I only have minor comments for the authors:

  1. The impact of HLA-G alleles in the vertical transmission of several viral infections including HIV and HCV has been reported in many previous studies. Authors should mention that in their introduction. Additionally, I recommend discussing if there is a common HLA-G feature that contributes to the vertical transmission of different viruses by comparing the current study findings with other studied viruses’ reports.

Thank you for this comment. The information has been added to the introduction section. Moreover, discussion about HLA-G risk alleles and vertical transmission of other viral infections has been added to the Discussion section.

  1. I recommend adding an introductory sentence in the first paragraph of the discussion instead of starting the discussion with (According to our results, .....).

Thank you for this comment. The introduction sentence has been added.

Round 2

Reviewer 1 Report

Dear Editor and Authors:

 Thank you for providing the point-to-point response.

The authors have carefully and patiently corrected and answered the comments and questions. The manuscript looks perfect now, and the reviewer suggests it be accepted for publication in this journal.

Please feel free to contact me if you have further requests or concerns.

Thank you for reading.

Minor revision is needed.